# Fast Rates for Bandit PAC Multiclass Classification

**Liad Erez**
Tel-Aviv University
liaderez@mail.tau.ac.il

**Alon Cohen**
Tel-Aviv University
Google Research
alonco@tauex.tau.ac.il

**Tomer Koren**
Tel-Aviv University
Google Research
tkoren@tauex.tau.ac.il

**Yishay Mansour**
Tel-Aviv University
Google Research
mansour.yishay@gmail.com

**Shay Moran**
Technion
Google Research
shaymoran1@gmail.com

## Abstract

We study multiclass PAC learning with bandit feedback, where inputs are classified into one of $K$ possible labels and feedback is limited to whether or not the predicted labels are correct. Our main contribution is in designing a novel learning algorithm for the agnostic $(\varepsilon, \delta)$-PAC version of the problem, with sample complexity of $O\big((\text{poly}(K) + 1/\varepsilon^2) \log(|\mathcal{H}|/\delta)\big)$ for any finite hypothesis class $\mathcal{H}$. In terms of the leading dependence on $\varepsilon$, this improves upon existing bounds for the problem, that are of the form $O(K/\varepsilon^2)$. We also provide an extension of this result to general classes and establish similar sample complexity bounds in which $\log |\mathcal{H}|$ is replaced by the Natarajan dimension. This matches the optimal rate in the full-information version of the problem and resolves an open question studied by Daniely, Sabato, Ben-David, and Shalev-Shwartz (2011) who demonstrated that the multiplicative price of bandit feedback in realizable PAC learning is $\Theta(K)$. We complement this by revealing a stark contrast with the agnostic case, where the price of bandit feedback is only $O(1)$ as $\varepsilon \to 0$. Our algorithm utilizes a stochastic optimization technique to minimize a log-barrier potential based on Frank-Wolfe updates for computing a low-variance exploration distribution over the hypotheses, and is made computationally efficient provided access to an ERM oracle over $\mathcal{H}$.

## 1 Introduction

Multiclass classification is a fundamental learning problem in which a learner is tasked with classifying objects into one of $K$ possible labels. In *bandit multiclass classification* [18], upon making a prediction, the learner does not observe the true label, but only whether or not the prediction was correct. As a concrete example, consider a the task of classifying images, say, from the ImageNet dataset, with the number of labels $K$ being several thousands. After the learner predicts a label for a particular image, both the image and the prediction are shown to a human rater who is asked if the prediction is correct or not, after which the answer is revealed to the learner. Thus, the learner faces a bandit multiclass classification instance as the true label is not revealed in case the prediction was deemed incorrect by the rater.

Much of previous work on the foundations of bandit multiclass classification focused on the *online* setting [18, 10, 9, 21, 23, 13], where the goal of the learner is to minimize the *regret* compared to a given *hypothesis class* $\mathcal{H}$, namely, the learner's total number of correct predictions throughout the learning process compared to that of the best fixed hypothesis from $\mathcal{H}$. A central line of work focused on studying the properties of $\mathcal{H}$ that allow for sublinear regret in this context, and on char-

38th Conference on Neural Information Processing Systems (NeurIPS 2024).

acterizing the achievable regret rates in terms of $K$, $|\mathcal{H}|$ and the number of prediction rounds $T$. For instance, Auer et al. [4] show that for any finite $\mathcal{H}$ one can obtain a regret bound of $O(\sqrt{KT \log |\mathcal{H}|})$, by casting the classification problem as a (contextual) $K$-armed bandit problem. Daniely and Helbertal [9] demonstrate that the dependence on $\log |\mathcal{H}|$ can be replaced by the Littlestone dimension of $\mathcal{H}$, which may potentially be smaller than $\log |\mathcal{H}|$ and, in particular, may be finite even when $\mathcal{H}$ is infinite. Very recently, Erez et al. [13] establish a characterization of the optimal regret rates for finite hypothesis classes and show that it is of the form $\Theta(\min\{\sqrt{KT \log |\mathcal{H}|}\}, |\mathcal{H}| + \sqrt{T})$, which is tight even when the labeled examples are drawn i.i.d. from a fixed distribution.

Here we focus on a different, yet closely related version of multiclass classification, viewed as a learning problem in a PAC framework [25]. In this setting, the labeled examples are drawn from a fixed unknown distribution, and the learner's goal is to ultimately output a prediction rule which performs well, over this distribution, relative to the best hypothesis from the underlying class $\mathcal{H}$. This problem has mainly been studied in the analogous full-information setting, that is, when the learner as access to a training set of i.i.d. examples along with their true labels, where the number of samples required to learn an $\varepsilon$-optimal hypothesis was shown to be $O((1/\varepsilon^2) \log(|\mathcal{H}|/\delta))$ for finite classes [22, 5, 10, 6]. In the bandit case, however—namely where the learner may repeatedly predict labels of drawn examples and obtain feedback only on whether the prediction was correct or not—a comprehensive understanding of the achievable sample complexity rates is still missing.

On the surface, the PAC bandit multiclass problem might be deemed trivial: a straightforward approach of uniformly approximating the losses of all hypotheses in $\mathcal{H}$ by drawing i.i.d. examples and predicting labels uniformly at random, already gives rise to $O((K/\varepsilon^2) \log |\mathcal{H}|)$ sample complexity which appears to be optimal due to the bandit feedback. Furthermore, the simple underlying algorithm can be implemented efficiently, provided that empirical risk minimization (ERM) can be carried out efficiently over the hypothesis class. However, this view regards multiclass classification as a generic $K$-armed (contextual) bandit problem and neglects a crucial aspect of the setting: each example has a single correct label, as opposed to a more general scenario where each label may be associated with its own loss, independently of other labels. And indeed, no non-trivial lower bounds can be found in the existing literature for this problem.[1]

The following questions thus remain:

(1) *What is the optimal sample complexity of the bandit multiclass setting? In particular, can one improve upon the prototypical $K/\varepsilon^2$ rate, representative of bandit problems?*

(2) *Can this sample complexity be attained by an efficient (polynomial time) algorithm, whenever ERM can be computed efficiently over the underlying hypothesis class?*

## 1.1 Summary of contributions

In this work, we address the questions above and establish a nearly-tight characterization of the achievable sample complexity in the bandit multiclass problem, along with an efficient algorithm. Our main contributions are summarized as follows.

(i) For a finite hypothesis class $\mathcal{H}$, we give an algorithm with sample complexity of $O((\text{poly}(K) + 1/\varepsilon^2) \log(|H|/\delta))$ for producing an $\varepsilon$-optimal classifier with probability at least $1 - \delta$; see Theorem 1 in Section 3. Further, our algorithm is a proper learner and can be implemented efficiently provided a (weighted) ERM oracle for the class $\mathcal{H}$. In terms of the leading dependence on $\varepsilon$, this bound significantly improves over the previously mentioned $O(K/\varepsilon^2)$ bound, and matches the optimal rate in the full-information version of the problem.

(ii) For more general, possibly infinite hypothesis classes $\mathcal{H}$ with finite Natarajan dimension $d$, we establish a generalized sample complexity bound of $O((\text{poly}(K) + 1/\varepsilon^2)d \log(1/\delta))$; see Theorem 2 in Section 4. This improves over the previous $O(Kd/\varepsilon^2)$ bound due to Daniely, Sabato, Ben-David, and Shalev-Shwartz [10], and matches the known rate in the full-information case [22] up to logarithmic factors for sufficiently small $\varepsilon$.

These results bear some interesting consequences. First, and perhaps most surprisingly, they imply that there is *no additional price for bandit feedback* in PAC multiclass classification, as $\varepsilon \to 0$;

---

[1]We note that Daniely and Helbertal [9] do provide a stochastic lower bound construction, but it pertains to multi-label multiclass rather to the standard single-label setting we consider here.

namely, that the ratio between the optimal sample complexity rates in the bandit and the full-information settings is $\Theta(1)$, and not $\Theta(K)$ as one might expect. Indeed, the latter is often the multiplicative price of bandit information in a multitude of scenarios [see, e.g., 19], and further, it is the tight price in the realizable case of bandit multiclass [10]. This phenomenon occurs already in the case of a finite hypothesis class, and extends naturally to general Natarajan classes.

Second, the results reveal an unexpected gap between the attainable bounds in the online (i.e., regret minimization) and the PAC settings of the bandit multiclass problem. A recognized trademark of online learning is that online-to-batch conversions of regret bounds very often give sharp sample complexity rates in the i.i.d. PAC setting [see e.g., 8, 7, 17]. In bandit multiclass classification, however, we exhibit a stark separation between the two settings, where for sufficiently large hypothesis classes an online-to-batch conversion of the optimal regret rate results with sample complexity $\widetilde{\Theta}(K/\varepsilon^2)$ [13], whereas the rate we establish here is $\widetilde{O}(1/\varepsilon^2 + \mathrm{poly}(K))$.[2]

The core novelty in our algorithmic approach is a stochastic optimization technique for efficiently recovering a low-variance exploration distribution over the hypotheses in $\mathcal{H}$, with variance $O(1)$ rather than the $O(K)$ obtained by simple uniform exploration of labels. We show that through minimization of a stochastic objective akin to a log-barrier (convex) potential over the induced label probabilities, such a distribution can be computed in a sample-efficient way and in turn be used to uniformly estimate the losses of all hypotheses in $\mathcal{H}$ via importance sampling. Moreover, we demonstrate how this stochastic optimization problem can be solved efficiently using a stochastic Frank-Wolfe method. More details about the algorithmic ideas and an overview of the analysis are given in Section 3.

## 1.2  Additional related work

**Bandit multiclass classification.**  In agnostic online multiclass classification with bandit feedback, regret bounds of the form $O(\sqrt{KT \log |\mathcal{H}|})$ can be obtained by viewing the problem as an instance of contextual multi-armed bandits [4]; this bound has been recently improved by [13] to $\Theta(\min\{\sqrt{KT \log |\mathcal{H}|}, |\mathcal{H}| + \sqrt{T}\})$ for the classification setting. [9] show how to replace the $\log |\mathcal{H}|$ dependence in the first bound by the Littlestone dimension of $\mathcal{H}$, and [23] show how $K$ can be replaced with a refined quantity which encapsulates the effective number of labels. Additional refinements in the realizable setting include the *Bandit Littlestone dimension* which provides a characterization of the optimal mistake bound for deterministic learning algorithms [10].

**Contextual bandits.**  The PAC objective of the bandit multiclass classification problem can be seen as a special case of the problem of identifying an approximately optimal policy in contextual multi-armed bandits. In the more general contextual bandit framework, sample complexity lower bounds of $\widetilde{\Omega}(K/\varepsilon^2)$ are known [4], with regret upper bounds of the form $\widetilde{O}(\sqrt{KT})$ obtained in several previous works [4, 12, 1]. Reducing our problem to contextual bandits, however, ignores the special structure exhibited by the reward function in the classification setting, namely their sparsity (see [13] for a more detailed comparison), and indeed we establish improved sample complexity rates in this special case. In a bit more detail, in the works of [12, 1], one of the main technical ideas is to compute a distribution over policies which induces a reward estimator whose variance is bounded, uniformly over the policies, by $O(K)$ (it is actually nontrivial to show that such a distribution even exists). Our approach involves similar ideas in the sense that we also aim to compute such a low-variance exploration distribution over the hypotheses, but the crucial difference is that in the classification setting, the sparse nature of the rewards allows us to uniformly bound the variance by $O(1)$ rather than $O(K)$. Li et al. [20] also consider the PAC objective for contextual bandits and establish a gap dependent sample complexity bound which never exceeds $K/\varepsilon^2$, but to our understanding does not imply an improved bound for sparse rewards (or single-label classification), as even for the multi-armed bandit variant, in order to obtain improved sample complexity bounds which are instance dependent, one must leverage the variances of the arm rewards in addition to the gaps.

---

[2] A similar separation has been observed before, e.g., in the context of general online/stochastic convex optimization [24].

## 2 Problem Setup

**Bandit multiclass classification.** We consider a learning setting in which a learner is tasked with classifying objects from a set of examples $\mathcal{X}$ with a single label from a set of $K$ possible labels $\mathcal{Y} = \{1, \ldots, K\}$. A stochastic multiclass classification instance is specified by a hypothesis class $\mathcal{H} \subseteq \{\mathcal{X} \to \mathcal{Y}\}$ and a joint distribution $\mathcal{D}$ over example-label pairs over $\mathcal{X} \times \mathcal{Y}$. We focus on finite hypothesis classes and denote $N \triangleq |\mathcal{H}|$. In the bandit setup, the learner interacts with the environment in an iterative manner according to the following protocol, over $i = 1, 2, \ldots$:

(i) The environment generates a pair $(x_i, y_i) \sim \mathcal{D}$ and the example $x_i$ is revealed to the learner;
(ii) The learner predicts a label $\widehat{y}_i \in \mathcal{Y}$;
(iii) The learner observes whether or not the classification of $x_i$ is correct, namely $\mathbb{1}\{\widehat{y}_i = y_i\}$.[3]

**Agnostic PAC model.** In the PAC version of the problem, given parameters $\varepsilon, \delta > 0$ the goal of the learner is to produce a hypothesis $\hat{h} : \mathcal{X} \to \mathcal{Y}$,[4] such that with probability at least $1 - \delta$ (over the randomness of the environment as well as any internal randomization of the algorithm):

$$L_{\mathcal{D}}(\hat{h}) - L_{\mathcal{D}}(h^\star) \leq \varepsilon,$$

where here $L_{\mathcal{D}}(h) \triangleq \Pr[h(x) \neq y]$ is the expected zero-one loss of $h$ with respect to $(x, y) \sim \mathcal{D}$, and $h^\star \triangleq \operatorname{argmin}_{h \in \mathcal{H}} L_{\mathcal{D}}(h)$ is the best hypothesis in $\mathcal{H}$. That is, the learner needs to identify a hypothesis which is $\varepsilon$-optimal in $\mathcal{H}$ with respect to the expected zero-one loss, with probability at least $1 - \delta$. In this model, the learner's performance is measured in terms of *sample complexity*, which is the number of interaction rounds with the environment as a function of $\varepsilon, \delta$ required for satisfying the guarantee stated above.

**Weighted ERM oracle.** For our computational results, we will assume a weighted empirical risk minimization (ERM) oracle access to the hypothesis class $\mathcal{H}$. In more detail, we assume access to an oracle, denoted $\mathcal{O}_{\mathcal{H}}$, defined as follows: given a sequence of examples, labels and weights $(x_1, y_1, \alpha_1), \ldots, (x_t, y_t, \alpha_t) \in \mathcal{X} \times \mathcal{Y} \times \mathbb{R}$ as input, the oracle $\mathcal{O}_{\mathcal{H}}$ returns

$$\operatorname*{argmin}_{h \in \mathcal{H}} \sum_{s=1}^{t} \alpha_s \mathbb{1}\{h(x_s) \neq y_s\}. \tag{1}$$

We remark that this is a version of the argmin oracle often considered in the more general contextual bandit setting [e.g., 12, 1], specialized for the classification setting we focus on here. For our runtime results, we will assume that each call to $\mathcal{O}_{\mathcal{H}}$ takes $O(1)$ time.

**Additional notation.** We denote by $\Delta_N \triangleq \{P \in \mathbb{R}_+^N \mid \sum_{i=1}^{N} P_i = 1\}$ the $N$-dimensional *simplex* which corresponds to the set of all probability distributions over $\mathcal{H}$. Given $P \in \Delta_N$ and $h \in \mathcal{H}$ we use the notation $P(h)$ to denote the probability assigned to $h$ by the probability vector $P$. Given an example-label pair $(x, y) \in \mathcal{X} \times \mathcal{Y}$ we define the binary vector $r_{x,y} \in \{0, 1\}^N$ by

$$r_{x,y}(h) \triangleq \mathbb{1}\{h(x) = y\} \quad \forall h \in \mathcal{H},$$

that is, $r_{x,y}(h)$ is the zero-one reward of the hypothesis $h$ on the pair $(x, y)$. Given $P \in \Delta_N$ and $(x, y) \in \mathcal{X} \times \mathcal{Y}$ we denote the probability of choosing the label $y$ when sampling a hypothesis from $P$ on $x$ by

$$W_{x,y}(P) \triangleq \sum_{h = \mathcal{H}} P(h) \mathbb{1}\{h(x) = y\} = P \cdot r_{x,y},$$

and for $\gamma \in (0, 1)$ we let $W_{x,y}^\gamma(P) \triangleq (1 - \gamma) W_{x,y}(P) + \gamma/K$, which corresponds to mixing the distribution $W_{x,y}(P)$ with a uniform distribution over labels with weight factor $\gamma$.

---

[3]Note that in the bandit setting, the learner does *not* observe the true label $y_i$ directly.
[4]In this model we allow for *improper* learners, that is, the output hypothesis may not be a member of $\mathcal{H}$; we emphasize, however, that our main algorithm below is a *proper* learner, namely it returns a hypothesis $\hat{h} \in \mathcal{H}$.

**Algorithm 1** Bandit PAC Multiclass Classification via Log Barrier Stochastic Optimization

---

Parameters: $M_1, M_2, \gamma \in (0, \frac{1}{2}]$.
Phase 1:
Initialize $S \leftarrow \emptyset$.
**while** $|S| < M_1$ **do**
    Environment generates $(x, y) \sim \mathcal{D}$, algorithm receives $x$.
    Predict $\hat{y}$ uniformly at random from $\mathcal{Y}$ and receive feedback $\mathbb{1}\{\hat{y} = y\}$.
    Update $S \leftarrow S \cup \{(x, y)\}$ if $y = \hat{y}$, otherwise $S$ is unchanged.
**end while**
Solve the stochastic optimization problem defined in Eq. (2) up to an additive error of $\mu = \gamma^2/2K^2$ using the dataset $S$. Let $\hat{P} \in \Delta_N$ be its output.
Phase 2:
**for** $i = 1, \ldots, M_2$ **do**
    Environment generates $(x_i, y_i) \sim \mathcal{D}$, algorithm receives $x_i$.
    With prob. $\gamma$, pick $\hat{y}_i \in \mathcal{Y}$ uniformly at random; otherwise sample $h_i \sim \hat{P}$ and set $\hat{y}_i = h_i(x_i)$.
    Predict $\hat{y}_i$ and receive feedback $\mathbb{1}\{\hat{y}_i = y_i\}$.
**end for**
Return:

$$\hat{h} = O_{\mathcal{H}}\Big(\{(x_i, \hat{y}_i, \alpha_i)\}_{i=1}^{M_2}\Big),$$

where $\alpha_i = \mathbb{1}\{y_i = \hat{y}_i\}/W_{\hat{P}}^{\gamma}(x_i, \hat{y}_i)$.

---

# 3 Algorithm and Analysis

In this section we present and analyze our main contribution: an efficient bandit multiclass classification algorithm, detailed in Algorithm 1, which will be shown to satisfy the following agnostic PAC guarantee.

**Theorem 1.** *If we set* $\gamma = \frac{1}{2}$, $M_1 = \Theta(K^8 \log(N/\delta))$, $M_2 = \Theta\big(\log(N/\delta)\big(K/\varepsilon + 1/\varepsilon^2\big)\big)$ *and use Algorithm 2 to solve the optimization problem defined in Eq. (2) for* $T = \Theta((K^4/\gamma^4)\sqrt{\log(N/\delta)})$ *rounds with step sizes* $\eta_t = 1/t$ *and batch sizes* $b_t = (\gamma/2K)^2 t^2$ *for* $t \in \{2^k - 1\}_{k \geq 1}$ *and* $b_t = t$ *otherwise; then with probability at least* $1 - \delta$ *Algorithm 1 outputs* $\hat{h} \in \mathcal{H}$ *with* $L_{\mathcal{D}}(\hat{h}) - L_{\mathcal{D}}(h^\star) \leq \varepsilon$, *using a total sample complexity of*

$$O\left(\left(K^9 + \frac{1}{\varepsilon^2}\right)\log\frac{N}{\delta}\right).$$

*Furthermore, Algorithm 1 makes a total number of* $O\big(K^4\sqrt{\log(N/\delta)}\big)$ *calls to the weighted ERM oracle* $O_{\mathcal{H}}$, *and runs in total time polynomial in* $K$, $1/\varepsilon$ *and* $\log(N/\delta)$.

Algorithm 1 operates in two phases. In the first phase, we construct a dataset $S$ of $M_1 = \text{poly}(K)\log(N/\delta)$ i.i.d. samples from $\mathcal{D}$ by predicting labels uniformly at random and taking into $S$ the samples for which the correct label is predicted (and is thus known). We then feed these samples to a stochastic optimization scheme which finds an approximate solution to the following stochastic optimization problem:

$$\min \quad \Phi(P) \triangleq \mathbb{E}_{(x,y)\sim\mathcal{D}}[\phi(P; x, y)], \quad \text{where} \quad \phi(P; x, y) \triangleq -\log\big(W_{x,y}^{\gamma}(P)\big) \qquad (2)$$
$$\text{s.t.} \quad P \in \Delta_N.$$

The approximate solution, $\hat{P} \in \Delta_N$ will be shown to satisfy certain low-variance properties, which will allow us to repeatedly sample from $\hat{P}$ for $M_2 = O\big(\big(K/\varepsilon + 1/\varepsilon^2\big)\log(N/\delta)\big)$ and estimate the loss of hypotheses in $\mathcal{H}$ rounds such that we are guaranteed to find an approximately optimal policy with high probability. We again emphasize that our algorithm is a proper learner in the sense that the returned hypothesis $\hat{h}$ is a member of the underlying class $\mathcal{H}$.

## 3.1 Overview of analysis

We next outline the details and intuition behind the implementation of Algorithm 1.

**Low-variance exploration distribution.** The main goal of the first phase of Algorithm 1 is to compute an exploration distribution $\hat{P} \in \Delta_N$ with the property that with probability at least $1 - \delta/2$, the following holds:

$$\forall h \in \mathcal{H} : \qquad \mathbb{E}_{(x,y) \sim \mathcal{D}} \left[ \frac{r_{x,y}(h)}{W_{x,y}^{\gamma}(\hat{P})} \right] \leq C, \tag{3}$$

where $\gamma$ is some predefined parameter and $C$ is an absolute constant, which crucially does not depend on $K$. The intuition behind this property is that the quantity of the left-hand side of Eq. (3) constitutes an upper bound on the variance of the random variable

$$\frac{r_{x,y}(h) \mathbb{1}\{y = \hat{y}\}}{W_{x,y}^{\gamma}(\hat{P})},$$

which is an unbiased estimator of the expected reward of the hypothesis $h$, that is, of $\mathbb{1}\{h(x) = y\}$ if the label $\hat{y}$ is chosen according to an hypothesis drawn from $\hat{P}$ and $(x, y)$ is drawn from $\mathcal{D}$. Thus, we think of Eq. (3) as a property which guarantees that the exploration distribution $\hat{P}$ can be used to estimate rewards, *uniformly* for all hypotheses in $\mathcal{H}$, with low (constant) variance. This in turn allows us to make use of variance-sensitive concentration bounds, namely Bernstein's inequality, in order to accurately approximate the optimal policy using a small number of samples (this is done in the second phase of the algorithm, described below).

**Controlling variance via stochastic optimization.** In some more detail, our approach for computing such a low-variance exploration distribution $\hat{P}$ is via approximately solving the stochastic convex optimization problem defined in Eq. (2). The reason for the choice of $\Phi$ as a convex potential to be minimized is the fact that its gradient is, up to a constant factor, given by

$$\forall h \in \mathcal{H} : \qquad (\nabla \Phi(P))_h \approx \mathbb{E}_{(x,y) \sim \mathcal{D}} \left[ \frac{r_{x,y}(h)}{W_{x,y}^{\gamma}(P)} \right], \tag{4}$$

so that finding a low-variance exploration distribution amounts to computing $\hat{P} \in \Delta_N$ for which $\nabla \Phi(\hat{P})$ is bounded in $L_\infty$ norm. Indeed, we show that the optimal solution to the optimization problem given in Eq. (2) satisfies such a property. This, together with the properties of $\Phi$ as a self-concordant function allows finding such $\hat{P}$ by approximately minimizing $\Phi$ over the simplex: the self-concordance of $\Phi$ acts as a "restricted strong convexity" property, which roughly implies that approximate minimizers of $\Phi$ must also have small (low norm) gradients.

**Efficient optimization via Stochastic Frank-Wolfe.** In order to compute an approximate minimizer of the convex potential $\Phi$ defined in Eq. (2), we employ a stochastic optimization procedure, formally described in Algorithm 2, which is based on a stochastic version of the Frank-Wolfe (FW) algorithm [15] with SPIDER gradient estimates [14]. The reason for choosing a FW based approach, is that it allows for efficient optimization of $\Phi$ in $N$-dimensional space, with runtime essentially independent of $N$, by exploiting the weighted ERM oracle at our disposal. Furthermore, the FW algorithm, when performed over the simplex, generates iterates $P_1, P_2, \ldots$ such that $P_t$ is supported on at most $t$ coordinates (provided that $P_1$ is initialized at an arbitrary vertex of the simplex), allowing us to maintain a succinct representation of the FW iterates—again, essentially independently of the ambient dimension $N$. Additionally, we remark that while existing analyses (e.g. [29]) of the stochastic FW algorithm rely on smoothness of the objective with respect to the $L_2$-norm, our objective of interest, namely $\Phi$, is not smooth in this classical sense, however it is smooth with respect to the $L_1$ norm. Therefore, we crucially rely on a different analysis of the FW algorithm with SPIDER gradient estimates, presented in Appendix C, which accommodates smoothness with respect to the $L_1$ norm.

**Final exploration phase.** The second phase of Algorithm 1 is more straightforward, where we repeatedly predict labels using i.i.d. samples from a distribution which mixes the distribution over labels induced by $\hat{P}$ (the exploration distribution computed in phase 1) with a uniform distribution over $\mathcal{Y}$. Using Bernstein's inequality and the uniform low-variance property of $\hat{P}$, we show that $M_2 = O\left( \left( K/\varepsilon + 1/\varepsilon^2 \right) \log(N/\delta) \right)$ samples suffice in order to ultimately output an hypothesis $\hat{h} \in \mathcal{H}$ being $\varepsilon$-optimal with probability at least $1 - \delta/2$. With this in hand, the desired PAC guarantee follows by a union bound over the failure probabilities of the two phases of the algorithm.

## 3.2 Stochastic Frank-Wolfe with SPIDER Gradient Estimates

Next, we present the stochastic FW procedure used in our algorithm as subprocedure; see Algorithm 2. It is essentially the SPIDER FW algorithm of Yurtsever et al. [29], specialized for solving the stochastic optimization problem defined in Eq. (2). We remark that the algorithm makes calls to a linear optimization oracle over the simplex denoted by $\mathsf{LOO}$, that given an input $g \in \mathbb{R}^N$ computes $\mathsf{LOO}(g) = \operatorname{argmin}_{Q \in \Delta_N} Q \cdot g$. As we will show later, each of the calls Algorithm 2 makes to $\mathsf{LOO}$ can be implemented by a call to the weighted ERM oracle. We also remark that, as discussed before, existing analyses of the stochastic FW procedure with SPIDER gradient estimates relies on $L_2$ smoothness of the objective [29], and here we provide an analysis with respect to $L_1$ smoothness being crucial in our case. We defer further details about SPIDER FW and its analysis to Appendix C.

---

**Algorithm 2** Stochastic Frank-Wolfe with SPIDER gradient estimates

---

Parameters: Dataset $S \subseteq \mathcal{X} \times \mathcal{Y}$, step sizes $\{\eta_t\}_t$, batch sizes $\{b_t\}_t$.
Initialize $P_1 \in \Delta_N$ and initial gradient estimate $g_1 = 0$.
Let $\tau_k = 2^k - 1$ for $k = 1, 2, \ldots$.
**for** $t = 1, 2, \ldots$ **do**
    Draw $b_t$ fresh samples $(x_1, y_1), \ldots, (x_{b_t}, y_{b_t})$ from $S$;
    **if** $t \in \{\tau_k\}_{k \geq 1}$ **then**
      Set

$$g_t = \frac{1}{b_t} \sum_{i=1}^{b_t} \nabla \phi(P_t; x_i, y_i);$$

    **else**
      Set

$$g_t = g_{t-1} + \frac{1}{b_t} \sum_{i=1}^{b_t} (\nabla \phi(P_t; x_i, y_i) - \nabla \phi(P_{t-1}; x_i, y_i));$$

    **end if**
    Compute $Q_t = \mathsf{LOO}(g_t)$;
    Update $P_{t+1} = (1 - \eta_t) P_t + \eta_t Q_t$;
**end for**

---

## 3.3 Proof of Theorem 1

We now turn to formally prove Theorem 1. For clarity of notation, we henceforth use $\mathbb{E}[\cdot]$ instead of $\mathbb{E}_{(x,y) \sim \mathcal{D}}[\cdot]$ to denote an expectation of a random variable with respect to $\mathcal{D}$. We begin this section by analyzing the first phase of Algorithm 1. The following lemma, which is proven in Appendix A, characterizes the optimal solution to the stochastic optimization problem defined in Eq. (2), and shows that the gradient of $\Phi$ at that optimum is bounded by a constant in $L_\infty$ norm.

**Lemma 1.** *Suppose $\gamma \leq \frac{1}{2}$ and let $P_\star \in \operatorname{argmin}_{P \in \Delta_N} \Phi(P)$, where $\Phi : \Delta_N \to \mathbb{R}_+$ was defined in Eq. (2). Then for any $P \in \Delta_N$ we have*

$$\mathbb{E}\left[\frac{W_{x,y}^\gamma(P)}{W_{x,y}^\gamma(P_\star)}\right] \leq 1. \tag{5}$$

*In particular, letting $P$ be the delta distribution on some $h \in \mathcal{H}$:*

$$\mathbb{E}\left[\frac{r_{x,y}(h)}{W_{x,y}^\gamma(P_\star)}\right] \leq 2.$$

The next key lemma, whose proof is also in Appendix A ensures that a sufficiently approximate minimizer of $\Phi$ is also a point in which the gradient $\Phi$ is bounded in $L_\infty$-norm, establishing the property given in Eq. (3).

**Lemma 2.** *Suppose $\gamma \leq \frac{1}{2}$ and assume that for all $P_\star \in \Delta_N$ the following holds for $\hat{P}$ which was computed in phase 1 of Algorithm 1:*

$$\Phi(\hat{P}) - \Phi(P_\star) \leq \mu.$$

*Then,*

$$\|\nabla \Phi(\hat{P})\|_\infty \leq 2 + \sqrt{\frac{2\mu K^2}{\gamma^2}}.$$

*In particular, setting $\mu = \gamma^2/2K^2$ gives $\|\nabla \Phi(\hat{P})\|_\infty \leq 3$.*

The fact that the property given in Eq. (3) can be guaranteed with high probability using as few as $M_1 = \Theta(\text{poly}(K)\log(N/\delta))$ samples relies on the following lemma, also proven in Appendix A, which follows from the analysis of the stochastic Frank-Wolfe procedure (see Appendix C for the detailed analysis).

**Lemma 3.** *Suppose $\gamma \leq \frac{1}{2}$. If $M_1 = 58000\left(K^8/\gamma^8\right)\log(16N/\delta)$ and $T = 240\left(K^4/\gamma^4\right)\sqrt{\log(16N/\delta)}$, then with probability at least $1 - \delta/2$, for all $P_\star \in \Delta_N$ it holds that*

$$\Phi(\hat{P}) - \Phi(P_\star) \leq \frac{\gamma^2}{2K^2}.$$

We are now in position to prove Theorem 1 by analyzing the second phase of Algorithm 1. We make use of the guarantee of the first phase given in Lemma 2 with respect to the exploration distribution $\hat{P}$, which allows us to use Bernstein's concentration inequality in order to compute an $\varepsilon$-optimal hypothesis in high probability using only $\approx K/\varepsilon + 1/\varepsilon^2$ samples.

*Proof of Theorem 1.* We first show that we can in fact use Algorithm 2 as specified in the theorem's statement. That is, we prove that the calls to the linear optimization oracle, denoted by LOO in Algorithm 2 can be implemented using the weighted ERM oracle $O_\mathcal{H}$. Indeed, we first note that for any $g \in \mathbb{R}^N$ it holds that

$$\text{LOO}(g) = \underset{P \in \Delta_N}{\text{argmin}}\{P \cdot g\} = \underset{h \in \mathcal{H}}{\text{argmin}}\{g_h\},$$

so it suffices to show that this can be represented in the form given in Eq. (1) for the SPIDER gradient estimates used in Algorithm 2, which we denote by $g_t$. It is straightforward to see each $g_t$ is a linear combination of terms of the form $\nabla\phi(P, x, y)$ where $P$ is either $P_t$ or $P_{t-1}$. Thus, we show that for $g = \frac{1}{n}\sum_{i=1}^n \nabla\phi(P_t, x_i, y_i)$, LOO($g$) can be computed by a call to the weighted ERM oracle. Indeed,

$$\text{LOO}(g) = \underset{h \in \mathcal{H}}{\text{argmin}}\left\{-(1-\gamma)\frac{1}{n}\sum_{i=1}^n \frac{\mathbb{1}\{h(x_i) = y_i\}}{W_{x_i,y_i}^\gamma(P_t)}\right\} = \underset{h \in \mathcal{H}}{\text{argmin}} \sum_{i=1}^n \alpha_i \mathbb{1}\{h(x_i) \neq y_i\},$$

where $\alpha_i = (1-\gamma)/nW_{x_i,y_i}^\gamma(P_t)$. Now, assume that after phase 1, Algorithm 1 computed $\hat{P} \in \Delta_N$ with

$$\max_{h \in \mathcal{H}} \mathbb{E}\left[\frac{r_{x,y}(h)}{W_{x,y}^\gamma(\hat{P})}\right] \leq 3.$$

Fix some $h \in \mathcal{H}$. For $i \in \{1, \ldots, M_2\}$ define $X_i(h) = \frac{r_{x_i,y_i}(h)\mathbb{1}\{y_i=\hat{y}_i\}}{W_{x_i,y_i}^\gamma(\hat{P})}$. Note that $\mathbb{E}[X_i(h)] = \Pr[h(x) = y]$ and that $X_1(h), \ldots, X_{M_2}(h)$ are i.i.d. Additionally, by the guarantee of phase 1:

$$\text{Var}[X_i(h)] \leq \mathbb{E}[X_i(h)^2] = \mathbb{E}\left[\frac{r_{x_i,y_i}(h)\mathbb{1}\{y_i = \hat{y}_i\}}{(W_{x_i,y_i}^\gamma(\hat{P}))^2}\right] = \mathbb{E}\left[\frac{r_{x,y}(h)}{W_{x,y}^\gamma(\hat{P})}\right] \leq 3.$$

Define $\bar{X}(h) = \frac{1}{M_2}\sum_{i=1}^{M_2} X_i(h)$, and note that by the form of the weighted ERM oracle, $\hat{h} = \text{argmax}_{h \in \mathcal{H}} \bar{X}(h)$. Therefore, by an application of Bernstein's inequality (see e.g. [19], page 86) and a union bound, we have with probability at least $1 - \delta/2$ for all $h \in \mathcal{H}$:

$$|\bar{X}(h) - \Pr[h(x) = y]| \leq \sqrt{\frac{6\log(2N/\delta)}{M_2}} + \frac{2K\log(2N/\delta)}{\gamma M_2}.$$

Hence for $\hat{h}$, with probability at least $1 - \delta/2$ we have
$$L_{\mathcal{D}}(\hat{h}) - L_{\mathcal{D}}(h^{\star}) = \Pr[\hat{h}(x) \neq y] - \Pr[h^{\star}(x) \neq y]$$
$$= \Pr[h^{\star}(x) = y] - \Pr[\hat{h}(x) = y]$$
$$\leq \Pr[h_{\star}(x) = y] - \bar{X}(h^{\star}) + \bar{X}(\hat{h}) - \Pr[\hat{h}(x) = y]$$
$$\leq \sqrt{\frac{36 \log(2N/\delta)}{M_2}} + \frac{4K \log(2N/\delta)}{\gamma M_2}.$$

Choosing $M_2 \geq \max\{144 \log(2N/\delta)/\varepsilon^2, 8K \log(2N/\delta)/(\gamma\varepsilon)\}$ gives $L_{\mathcal{D}}(\hat{h}) - L_{\mathcal{D}}(h^{\star}) \leq \varepsilon$. Thus, using Lemma 2 and Lemma 3, we are only left with proving that with high probability, collecting $M_1$ samples for $S$ takes at most $O(K^9 \log(N/\delta))$ steps. This essentially follows from the fact that a binomial random variables is smaller than half of its expected value with very small probability. Formally, we use lemma F.4 of [11] to deduce that if $X \sim \mathrm{Bin}(4KM_1, 1/K)$ then with probability at least $1 - \delta$ it holds that $X \geq 2M_1 - \log(1/\delta) \geq M_1$, which means that $4KM_1$ trials suffice to guarantee that with probability at least $1 - \delta$, the dataset $S$ will contain at least $M_1$ samples. Thus, the proof of Theorem 1 is complete once we make the observation that $2K/\varepsilon \leq K^2 + 1/\varepsilon^2$ (using the AM-GM inequality) so that the $K/\varepsilon$ term is of lower order and can be dropped from the final bound. □

# 4 Extension to Natarajan Classes

In this section we extend our result for finite classes given in Theorem 1 to general, possibly infinite hypothesis classes $\mathcal{H}$ with finite *Natarajan dimension*. The Natarajan dimension is an extension of the VC dimension to the multiclass setting, defined as follows:

**Definition 1** (Natarajan dimension [22]). *The Natarajan dimension of a hypothesis class $\mathcal{H} \subseteq \mathcal{X} \to \mathcal{Y}$ is the largest integer $d$ for which there exist $d$ points $x_1, \ldots, x_d \in \mathcal{X}$ and $d$ pairs of distinct labels $\{y_{1,1}, y_{1,2}\}, \ldots, \{y_{d,1}, y_{d,2}\} \in \binom{|\mathcal{Y}|}{2}$ such that all $2^d$ sequences of the form $(x_1, y_1), \ldots, (x_d, y_d)$, with $y_i \in \{y_{i1}, y_{i2}\}$, are realizable by $\mathcal{H}$.*

Note that in the binary case, when $\mathcal{Y} = \{0, 1\}$, the Natarajan dimension reduces to the VC dimension. Our main result in this case effectively replaces the $\log|\mathcal{H}|$ term, relevant when $\mathcal{H}$ is finite, with the Natarajan dimension $d_N$ of $\mathcal{H}$.

**Theorem 2.** *A hypothesis class $\mathcal{H} : \mathcal{X} \to \mathcal{Y}$ is PAC learnable with bandit feedback if and only if (i) it has a finite Natarajan dimension, and (ii) there exists $K \in \mathbb{N}$ such that $|\{h(x) : h \in \mathcal{H}\}| \leq K$ for every $x \in \mathcal{X}$.*

*Furthermore, let $\mathcal{H} : \mathcal{X} \to \mathcal{Y}$ be a hypothesis class of finite Natarajan dimension $d_N$. Then there exists a bandit multiclass classification algorithm which outputs a hypothesis $\hat{h}$ with $L_{\mathcal{D}}(\hat{h}) - \inf_{h \in \mathcal{H}} L_{\mathcal{D}}(h) \leq \varepsilon$ with a sample complexity of $\widetilde{O}((K^9 + 1/\varepsilon^2)d_N \log(1/\delta))$.*

As discussed, this result improves the classical result of Daniely et al. [10], who provided an upper bound on the sample complexity of PAC learning with bandit feedback, given by $\widetilde{O}(Kd_N/\varepsilon^2)$, and left obtaining tighter bounds as an open question. In particular, for small target excess loss ($\varepsilon \to 0$), our bound eliminates the linear dependence on the number of labels $K$, thereby matching the $\widetilde{\Theta}(d_N/\varepsilon^2)$ bound from the full information setting [22].

Theorem 2 follows directly from the following technical result, which is proven in Appendix B, when put together with Theorem 1.

**Proposition 1.** *Assume that the sample complexity of PAC learning with bandit feedback a finite class of size $N$ over a label-space of size $K$ is at most $m(N, K, \varepsilon, \delta)$, where $\epsilon, \delta$ are the error and confidence parameters. Then, the sample complexity of learning an infinite class $\mathcal{H}$ is at most*
$$s + m(S, K, \epsilon/2, \delta/2),$$
*where*
$$s := O\left(\frac{d_N \ln(K) \ln(1/\epsilon) + \ln(1/\delta)}{\epsilon}\right),$$
$$S := \sum_{i=0}^{d_N} \binom{s}{i} \binom{K}{2}^i \leq \left(\frac{es}{d_N}\right)^{d_N} K^{2d_N},$$

*and $d_N$ is the Natarajan dimension of $\mathcal{H}$.*

The proposition is proven by using the first $s$ examples to construct a finite discretization of size $S$ of the class $\mathcal{H}$, which is $(\epsilon/2)$-dense in $\mathcal{H}$ in the sense that every $h \in \mathcal{H}$ is $(\epsilon/2)$-close to some $h'$ in the discretization. Then, we apply our algorithm for finite classes on this finite discretization. Most of the proof is dedicated to establishing that the discretization is dense and to upper bounding its size. A similar result for discretizing binary classes, with the VC dimension replacing the Natarajan dimension, is well known. In the VC case, the proof is based on considering the class of all symmetric differences of hypotheses from $\mathcal{H}$ and analyzing the VC dimension of that class using standard techniques. In our proof, we follow a similar argument, but the analysis of the VC dimension of the symmetric difference class, which remains binary even in the multiclass setting, is more nuanced.

*Proof of Theorem 2.* For the first part of the theorem, note that Item (i) is clearly necessary for learnability, as a finite Natarajan dimension is required even in the case of full information feedback. Item (ii) is also necessary: if there exists a point $x$ such that $\{h(x) : h \in \mathcal{H}\}$ contains more than $K$ distinct labels, then in the realizable case, the sample complexity with bandit feedback has a lower bound of $\Omega(K)$, provided $K$ is a sufficiently large constant. This follows by setting the target distribution to assign probability 1 to a single example $(x, y)$, where $y$ is drawn uniformly in advance from $K$ labels in $\{h(x) : h \in \mathcal{H}\}$. An elementary coupon collector argument yields a lower bound of $K/2$: indeed, if only $m \leq K/2$ examples are drawn, then the learner does not correctly guess the label during training with probability at least $1/2$. Conditioned on this event, the population loss is $\geq 1/2$ (in fact, $\geq \frac{1}{K/2}$), leading to a lower bound of $1/2 \cdot 1/2 = 1/4$ on the population error.

For sufficiency, note that if Item (ii) holds, the class $\mathcal{H}$ effectively reduces to a class over $K$ total labels and is therefore learnable when its Natarajan dimension is finite, as follows from [9].

For the second part of the theorem, note that by Theorem 1, we have an upper bound on the sample complexity for PAC learning with bandit feedback over finite classes of size $N$ of

$$m(N, K, \varepsilon, \delta) = O\Big((K^9 + 1/\varepsilon^2) \log(N/\delta)\Big).$$

Thus, the proof follows immediately from Proposition 1 together with the fact that:

$$\log\left(\frac{S}{\delta}\right) \leq d_N \log\left(\frac{es}{d_N}\right) + 2d_N \log K + \log \frac{1}{\delta},$$

where $s$ and $S$ are defined in the statement of Proposition 1 (note that $\log(s)$ only contributes logarithmic terms to the overall bound).

$\square$

# 5 Conclusion and Open Problems

In this work, we establish a nearly-optimal sample complexity of $\widetilde{O}((\text{poly}(K)+1/\varepsilon^2) \log(|H|/\delta))$ for the bandit multiclass classification problem and design an efficient algorithm achieving this bound. While the dominant term in our sample complexity bound (which depends on $\varepsilon$) is optimal, our bound exhibits a $K^9$ additive dependence on the size of the label space. We conjecture that the optimal bound is of the form $O(K/\varepsilon + 1/\varepsilon^2)$, for which a matching lower bound can be shown in a straightforward manner. One possible approach to improve the dependence on $K$ is perhaps a more adaptive method which combines maximization of estimated rewards in tandem with lowering the variance of the sampling distribution gradually over time. We leave this as a very interesting question for future research.

# Acknowledgments

This project has received funding from the European Research Council (ERC) under the European Union's Horizon 2020 research and innovation program (grant agreements No. 101078075; 882396). Views and opinions expressed are however those of the author(s) only and do not necessarily reflect those of the European Union or the European Research Council. Neither the European Union nor the granting authority can be held responsible for them. This project has also received

funding from the Israel Science Foundation (ISF, grant numbers 2549/19; 3174/23), the Yandex Initiative for Machine Learning at Tel Aviv University, the Tel Aviv University Center for AI and Data Science (TAD), the Len Blavatnik and the Blavatnik Family foundation, and from the Adelis Foundation.

Shay Moran is a Robert J. Shillman Fellow; supported by ISF grant 1225/20, by BSF grant 2018385, by an Azrieli Faculty Fellowship, by Israel PBC-VATAT, by the Technion Center for Machine Learning and Intelligent Systems (MLIS), and by the European Union (ERC, GENERALIZATION, 101039692). Views and opinions expressed are however those of the author(s) only and do not necessarily reflect those of the European Union or the European Research Council Executive Agency. Neither the European Union nor the granting authority can be held responsible for them.

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

## A  Proofs for Section 3

*Proof of Lemma 1.* First note that the gradient of $\Phi(\cdot)$ is given by

$$\nabla\Phi(P) = \mathbb{E}\left[-\frac{(1-\gamma)r_{x,y}}{W_{x,y}^{\gamma}(P)}\right].$$

Thus, using a first-order optimality condition for $P_\star$, the following holds that for any $P \in \Delta_N$,

$$\nabla\Phi(P_\star) \cdot (P - P_\star) \geq 0,$$

which amounts to

$$\mathbb{E}\left[\frac{(1-\gamma)(P-P_\star)\cdot r_{x,y}}{W_{x,y}^\gamma(P)}\right] \le 0,$$

or equivalently,

$$\mathbb{E}\left[\frac{W_{x,y}^\gamma(P) - W_{x,y}^\gamma(P_\star)}{W_{x,y}^\gamma(P_\star)}\right] \le 0,$$

which rearranges to the first inequality to be proven. Letting $P$ be the delta distribution on some $h \in \mathcal{H}$, we have proven that

$$\mathbb{E}\left[(1-\gamma)\frac{r_{x,y}}{W_{x,y}^\gamma(P_\star)}\right] \le 1,$$

and the second inequality follows since $\gamma \le \frac{1}{2}$. $\qquad\square$

*Proof of Lemma 2.* Assume that $\hat{P}$ minimizes $\Phi$ up to an additive error of $\mu$, that is, the assumption given in the statement of the lemma. Now, note that for every $(x, y) \in \mathcal{X} \times \mathcal{Y}$, using the explicit form of $\phi(\cdot; x, y)$ and its gradient, it holds that:

$$\phi(\hat{P}; x, y) - \phi(P_\star; x, y) - \nabla\phi(P_\star; x, y)\cdot(\hat{P} - P_\star) = \omega(R_{x,y}),$$

where $\omega(z) = -\log z + z - 1$ and $R_{x,y} = W_{x,y}^\gamma(\hat{P})/W_{x,y}^\gamma(P_\star)$. Using the lower bound $\omega(z) \ge \frac{1}{2}\min\{(1-z)^2, (1-\frac{1}{z})^2\}$ (see Lemma 4 in Appendix A; this is where the self-concordance of $\phi$ comes in) we obtain that for all $h \in \mathcal{H}$,

$$\phi(\hat{P}; x, y) - \phi(P_\star; x, y) - \nabla\phi(P_\star; x, y)\cdot(\hat{P} - P_\star) \ge \frac{\gamma^2}{2K^2}\left(\frac{1}{W_{x,y}^\gamma(\hat{P})} - \frac{1}{W_{x,y}^\gamma(P_\star)}\right)^2$$

$$\ge \frac{\gamma^2}{2K^2}\left(\frac{r_{x,y}(h)}{W_{x,y}^\gamma(\hat{P})} - \frac{r_{x,y}(h)}{W_{x,y}^\gamma(P_\star)}\right)^2.$$

Taking expectation of the inequality over $(x, y) \sim \mathcal{D}$ while using a first-order optimality condition for $P_\star$ with respect to $\Phi$ and the fact that $\mathbb{E}[(\cdot)^2] \ge \mathbb{E}^2[\cdot]$, we obtain

$$\mu \ge \Phi(\hat{P}) - \Phi(P_\star) \ge \frac{\gamma^2}{2K^2}\left(\mathbb{E}\left[\frac{r_{x,y}(h)}{W_{x,y}^\gamma(\hat{P})}\right] - \mathbb{E}\left[\frac{r_{x,y}(h)}{W_{x,y}^\gamma(P_\star)}\right]\right)^2,$$

and rearranging we obtain that for all $h \in \mathcal{H}$,

$$\mathbb{E}\left[\frac{r_{x,y}(h)}{W_{x,y}^\gamma(\hat{P})}\right] \le \sqrt{\frac{2\mu K^2}{\gamma^2}} + \mathbb{E}\left[\frac{r_{x,y}(h)}{W_{x,y}^\gamma(P_\star)}\right],$$

so using Lemma 1 and the form of $\nabla\Phi$ we obtain the desired bound. $\qquad\square$

*Proof of Lemma 3.* In order to invoke Theorem 3 on $\Phi(\cdot)$, we prove that $\phi(\cdot; x, y)$ satisfies the required Lipschitz and smoothness properties with $G = K/\gamma$ and $\beta = K^2/\gamma^2$. Indeed, the gradient of $\phi$ is given by

$$\nabla\phi(P, x, y) = -(1-\gamma)\frac{r_{x,y}}{W_{x,y}^\gamma(P)},$$

and since $W_{x,y}^\gamma(P) \ge \gamma/K$ and $\|r_{x,y}\|_\infty \le 1$ we obtain the required Lipschitz property, that is, $\|\nabla\phi(P, x, y)\|_\infty \le G$. For $L_1$ smoothness, it suffices to show that for any $P, Q \in \Delta_N$ and any $(x, y) \in \mathcal{X} \times \mathcal{Y}$ it holds that

$$\|\nabla\phi(P, x, y) - \nabla\phi(Q, x, y)\|_\infty \le \beta\|P - Q\|_1.$$

Indeed,

$$\|\nabla\phi(P,x,y) - \nabla\phi(Q,x,y)\|_\infty = (1-\gamma)\max_{h\in\mathcal{H}}\left|\frac{r_{x,y}(h)}{W_{x,y}^\gamma(P)} - \frac{r_{x,y}(h)}{W_{x,y}^\gamma(Q)}\right|$$

$$\leq \left|\frac{W_{x,y}^\gamma(Q) - W_{x,y}^\gamma(P)}{W_{x,y}^\gamma(P)W_{x,y}^\gamma(Q)}\right|$$

$$= (1-\gamma)\left|\frac{r_{x,y}\cdot(Q-P)}{W_{x,y}^\gamma(P)W_{x,y}^\gamma(Q)}\right|$$

$$\leq \frac{K^2}{\gamma^2}\|P-Q\|_1,$$

where in the last step we used Hölder's inequality. Now, using Theorem 3, our choice of $T$ and $M_1$ for which the batch sizes $b_t$ defined in Theorem 3 satisfy $\sum_{t=1}^T b_t \leq M_1$, the proof is complete. $\square$

**Lemma 4.** *Let $\omega(z) = -\log z + z - 1$. It holds that:*

$$\omega(z) \geq \min\left\{\tfrac{1}{2}(1-z)^2, \tfrac{z}{2}\left(1-\tfrac{1}{z}\right)^2\right\}.$$

*The first lower bound is relevant for $z \leq 1$ and the second for $z \geq 1$.*

*Proof.* Note that $\omega'(z) = 1 - 1/z$ and $\omega(1) = 0$, hence $\omega(z) = \int_1^z (1-1/x)dx$. When $0 < z \leq 1$ we can bound $1 - 1/x \leq 1 - x$ for any $z \leq x \leq 1$. Therefore,

$$\omega(z) = \int_1^z (1/x - 1)dx \geq \int_1^z (x-1)dx = \tfrac{1}{2}(1-z)^2.$$

When $z \geq 1$, we can use the elementary inequality, for any $z \geq x \geq 1$:

$$1 - \frac{1}{x} \geq \frac{1}{2}\left(1 - \frac{1}{x^2}\right) = \frac{d}{dx}\left\{\frac{x}{2}\left(1 - \frac{1}{x}\right)^2\right\}$$

to bound

$$\omega(z) = \int_1^z (1-1/x)dx \geq \int_1^z \left(\tfrac{1}{2}(1-1/x^2)\right)dx = \tfrac{z}{2}(1-1/z)^2.$$

$\square$

# B  Proof of Proposition 1

*Proof of Proposition 1.* Consider the following learning rule: first sample $s$ examples $x_1,\ldots,x_s$ from the population (guess their label arbitrarily). Now for each pattern $\{(x_i,y_i)\}_{i=1}^s$ which is realizable by some function $h \in \mathcal{H}$, pick a representative $h \in \mathcal{H}$ such that $h(x_i) = y_i$ for all $i \leq s$. Define $\mathcal{H}_{\texttt{fin}}$ to be the set of all such representatives. The Sauer-Shelah-Perles Lemma for the Natarajan dimension [16] implies that

$$|\mathcal{H}_{\texttt{fin}}| \leq S.$$

Now, apply the assumed algorithm on the finite class $\mathcal{H}_{\texttt{fin}}$ with error and confidence parameters $\varepsilon/2, \delta/2$. Let $\hat{h}$ denote the hypothesis outputted by the algorithm. Thus, with probability at least $\delta/2$ the excess loss (or regret) of $\hat{h}$ with respect to $\mathcal{H}_{\texttt{fin}}$ is at most $\varepsilon/2$. A union bound combined with the following lemma imply that with probability at least $\delta$, the excess loss of $\hat{h}$ with respect to $\mathcal{H}$ is at most $\varepsilon$.

**Lemma 5.** *With probability at least $1 - \delta/2$ over the sampling of $x_1,\ldots x_s$, the finite class $\mathcal{H}_{\texttt{fin}}$ is an $(\varepsilon/2)$-cover for $\mathcal{H}$. That is, for every $h \in \mathcal{H}$ there exists $h' \in \mathcal{H}_{\texttt{fin}}$ such that the probability that $h'(x) \neq h(x)$ for a random point $x$ drawn from the population is at most $\varepsilon/2$.*

Indeed, by a union bound, with probability at least $1 - \delta$, both (i) the excess loss of $\hat{h}$ is at most $\epsilon/2$, and (ii) $\mathcal{H}_{\text{fin}}$ is an $(\epsilon/2)$-cover for $\mathcal{H}$. Hence, the excess loss of $\hat{h}$ with respect to the best hypothesis in $\mathcal{H}$ is at most $\epsilon/2 + \epsilon/2 = \epsilon$, as required. $\qquad\square$

Lemma 5 follows from a standard uniform convergence argument. In the case of binary labels $K = 2$, this lemma is known and has been used e.g. in [2]. The general case is derived below using a similar argument like in [2].

*Proof of Lemma 5.* We need to show that with probability at least $1 - \delta/2$ over $(x_1, y_1), \ldots, (x_s, y_s) \sim \mathcal{D}$, for every $h \in \mathcal{H}$, there exists $h' \in \mathcal{H}_{\text{fin}}$ such that $\Pr_{(x,y) \sim \mathcal{D}}[h(x) \neq h'(x)] \leq \epsilon/2$. For convenience, we use the notation $d(h, h') := \Pr_{(x,y) \sim \mathcal{D}}[h(x) \neq h'(x)]$.

Let $T = (x_1, \ldots, x_s)$ be the sequence of points in the random sample, and for a hypothesis $h$ let $h(T) = (h(x_1), \ldots, h(x_s))$. By construction, for every $h \in \mathcal{H}$, there exists $h' \in \mathcal{H}_{\text{fin}}$ such that $h'(T) = h(T)$. We will show that $d(h, h') \leq \epsilon/2$.

Define the event

$$E = \big\{ \exists h_1, h_2 \in \mathcal{H} : \ d(h_1, h_2) > \epsilon/2 \ \text{and} \ h_1(T) = h_2(T) \big\}.$$

We prove that

$$\Pr[E] \leq 2 \left( \frac{2e\,s}{d_N} \right)^{2d_N} K^{4d_N}\, e^{-\epsilon\,s/4}. \tag{6}$$

Before we do so, we first show that this suffices to prove the lemma: indeed, if $d(h, h') > \epsilon/2$ for some $h \in \mathcal{H}$ and $h' \in \mathcal{H}_{\text{fin}}$ such that $h'(T) = h(T)$, then the event $E$ occurs because $\mathcal{H}_{\text{fin}} \subseteq \mathcal{H}$. Now, via standard manipulation, this bound is at most $\delta/2$ for some

$$s = O\left( \frac{d_N \ln(K) \ln(1/\epsilon) + \ln(1/\delta)}{\epsilon} \right),$$

which yields the desired bound and completes the proof.

It remains to prove (6). To do so, we use a standard VC-based uniform convergence bound on the class $\mathcal{H}_\Delta = \{\Delta_{h_1, h_2} : h_1, h_2 \in \mathcal{H}\}$, where

$$\Delta_{h_1, h_2}(x) = \begin{cases} 1 & h_1(x) \neq h_2(x), \\ 0 & h_1(x) = h_2(x). \end{cases}$$

Let $\mathcal{G}_\Delta$ denote the growth function of $\mathcal{H}_\Delta$; that is, for any number $m$,

$$\mathcal{G}_\Delta(m) = \max_{\{V : |V| = m\}} \big| \mathcal{H}_\Delta|_V \big|,$$

where $\mathcal{H}_\Delta|_V$ is the set of all restrictions (or projections) on $V$ of functions from $\mathcal{H}_\Delta$. Note that

$$\mathcal{G}_\Delta(m) \leq \left( \frac{2e\,s}{d_N} \right)^{2d_N} K^{4d_N}.$$

This follows from the fact that for any set $V$ of size $m$, we have $|\mathcal{H}_\Delta|_V| \leq |\mathcal{H}|_V|^2$, since every binary vector in $\mathcal{H}_\Delta|_V$ is determined by a pair of functions in $\mathcal{H}|_V$. Hence,

$$\mathcal{G}_\Delta(m) \leq \max_{|V| = m} \big| \mathcal{H}|_V \big|^2 \leq \left( \frac{2e\,s}{d_N} \right)^{2d_N} K^{4d_N},$$

where the last inequality follows from the extended Sauer's Lemma for Natarajan classes applied on $\mathcal{H}$ [16].

Now, by invoking a uniform convergence argument, we have

$$
\begin{aligned}
\Pr[E] &= \Pr[\exists h_1, h_2 \in \mathcal{H} : \ d(h_1, h_2) > \epsilon/2 \ \text{and} \ h_1(T) = h_2(T)] \\
&= \Pr[\exists b \in \mathcal{H}_\Delta : \ d(b, b_0) \geq \epsilon/2 \ \text{and} \ b(T) = b_0(T)] &&\text{(Here $b_0$ is the all-zero vector)} \\
&\leq \ 2\mathcal{G}_\Delta(2\,s)\, e^{-\epsilon, s/4} &&\text{(double-sample symmetrization argument)} \\
&\leq 2 \left( \frac{2e\,s}{d_N} \right)^{2d_N} K^{4d_N}\, e^{-\epsilon\,s/4}.
\end{aligned}
$$

The bound in the third line is non-trivial; it is known as the double-sample argument which was used by Vapnik and Chervonenkis in their seminal paper [26]. The same argument is used in virtually all VC-based uniform convergence bounds. This proves inequality (6) and completes the proof of the lemma. □

## C    Stochastic (SPIDER) Frank-Wolfe in L1 Norm

In this section we present and analyze the stochastic Frank-Wolfe (FW) method used as a sub-procedure in our main algorithm. Crucially, we require a specific variant of stochastic FW due to Yurtsever et al. [28], called SPIDER-FW, that employs mini-batching and variance-reduced for stochastic gradient estimation.

SPIDER FW has been shown to obtain the optimal $1/\varepsilon^2$ rate of convergence in a general stochastic setting that involves a convex and smooth objective [28]. However, the existing analysis pertain to the Euclidean case, whereas we crucially require an $L_1$–$L_\infty$ analysis. We provide such analysis here with a specialized argument for variance-reduction with respect to the $L_1$ norm. (We note that similar arguments have appeared in [3] in the context of differentially-private optimization.)

**Setup.**    The setup for this section is the following. We consider a stochastic optimization problem of the form

$$\text{minimize} \quad F(w) = \mathbb{E}[f(w, z)]$$
$$\text{s.t.} \quad w \in W,$$

where $W$ is a convex domain in $\mathbb{R}^d$ and the expectation is over $z$ drawn from an underlying distribution $\mathcal{D}$. In the context of this section, we think of $\mathcal{D}$ as being unknown but assume that i.i.d. samples $z_1, z_2, \ldots \sim \mathcal{D}$ are readily available.

We further make the following assumptions:

- We assume that $f$ convex, $\beta$-smooth and $G$-Lipschitz (in its first argument) with respect to the $L_1$ norm over $W$, that is,

$$f(u, z) \le f(v, z) + \nabla f(v, z) \cdot (u - v) + \frac{\beta}{2} \|u - v\|_1^2 \quad \forall z \in \mathcal{Z}, \forall u, v \in W,$$

and further, that $\|\nabla f(v, z)\|_\infty \le G$ for all $z \in Z$ and $v \in \mathcal{W}$;

- We assume that the feasible domain $W$ has $L_1$ diameter $\le D$, that is, $\|u - v\|_1 \le D$ for all $u, v \in W$. the algorithmic access to the set $W$ is through a linear optimization oracle (LOO), that given any vector $g \in \mathbb{R}^d$ computes

$$\mathsf{LOO}(g) = \underset{w \in W}{\text{argmin}} \, g \cdot w.$$

### C.1    Stochastic FW with SPIDER gradient estimates

The SPIDER FW algorithm is presented in Algorithm 3. The algorithm makes Frank-Wolfe type updates using "SPIDER" gradient estimates $g_t$, that are computed from the raw stochastic gradients. The SPIDER estimates work in mini-batches, but employ low-variance bias correction in order to update the estimates without resetting them entirely from round to round, thus saving in the mini-batch sizes.

The main result of this section is given in the following theorem.

**Theorem 3.** *Algorithm 3 with step sizes $\eta_t = 1/t$ and batch sizes $b_t$ defined as*

$$b_t = \begin{cases} (G/\beta D)^2 \, t^2 & \text{if } t \in \{\tau_k\}_{k \ge 1}; \\ t & \text{otherwise,} \end{cases} \tag{7}$$

*guarantees that, for all $t \ge 2$ and any $\delta > 0$:*

$$\mathbb{E}[F(w_t) - F(w^*)] \le \frac{25\beta D^2}{t} \sqrt{\log d} \,, \quad \text{and} \quad \Pr\left(F(w_t) - F(w^*) \le \frac{25\beta D^2}{t} \sqrt{\log \frac{dt}{\delta}}\right) \ge 1 - \delta \,.$$

*Consequently, for convergence to within $\epsilon > 0$ with probability at least $1 - \delta$, the algorithm requires $O((\beta D^2/\epsilon)\sqrt{\log(d/\delta)})$ calls to LOO and $O(((G^2 D^2 + \beta^2 D^4)/\epsilon^2) \log(d/\delta))$ stochastic gradient computations.*

**Algorithm 3** Stochastic Frank-Wolfe with SPIDER gradient estimates
---
Parameters: step sizes $\{\eta_t\}_t$, batch sizes $\{b_t\}_t$.
Initialize $w_1 \in \mathcal{W}$ and initial gradient estimate $g_1 = 0$.
Let $\tau_k = 2^k - 1$ for $k = 1, 2, \ldots$.
**for** $t = 1, 2, \ldots$ **do**
    Draw fresh $b_t$ samples $z_1, \ldots, z_{b_t}$ from $\mathcal{D}$;
    **if** $t \in \{\tau_k\}_{k \geq 1}$ **then**
        Set

$$g_t = \frac{1}{b_t} \sum_{i=1}^{b_t} \nabla f(w_t, z_i);$$

    **else**
        Set

$$g_t = g_{t-1} + \frac{1}{b_t} \sum_{i=1}^{b_t} (\nabla f(w_t, z_i) - \nabla f(w_{t-1}, z_i));$$

    **end if**
    Compute $v_t = \mathsf{LOO}(g_t)$;
    Update $w_{t+1} = (1 - \eta_t)w_t + \eta_t v_t$;
**end for**
---

## C.2   Analysis

To analyze Algorithm 3, we first record a generic convergence guarantee for stochastic FW template with gradient estimates. The template operates as follows; starting from an arbitrary initialization $w_1 \in W$, for $t = 1, \ldots, T$:

   (i) get gradient estimator $g_t$ at $w_t$;
   (ii) use $\mathsf{LOO}$ to compute $v_t = \operatorname{argmin}_{w \in W} g_t \cdot w$;
   (iii) update $w_{t+1} = (1 - \eta_t)w_t + \eta_t v_t$.

**Lemma 6.** *Set $\eta_t = 1/t$ for all $t$ and suppose that the gradient estimates satisfy, for some $c > 0$,[5]*

$$\forall\, t \geq 1: \qquad \mathbb{E}\|g_t - \nabla F(w_t)\|_\infty \leq c\eta_t.$$

*Then the FW iterations guarantees for all $t \geq 2$ and $w^* \in W$ that:*

$$\mathbb{E}[F(w_t) - F(w^*)] \leq \frac{\beta D^2 + 2cD}{t}.$$

*Similarly, if the estimates satisfy*

$$\forall\, t \geq 1: \qquad \Pr(\|g_t - \nabla F(w_t)\|_\infty > c_\delta \eta_t) \leq \delta.$$

*for some $\delta \in (0, 1)$ and $c_\delta > 0$, then for any $t \geq 2$ and $w^* \in W$, with probability at least $1 - t\delta$,*

$$F(w_t) - F(w^*) \leq \frac{\beta D^2 + 2c_\delta D}{t}.$$

*Proof.* First, using $\beta$-smoothness (with respect to $\|\cdot\|_1$) we have that

$$F(w_{t+1}) \leq F(w_t) + \nabla F(w_t) \cdot (w_{t+1} - w_t) + \tfrac{1}{2}\beta\|w_{t+1} - w_t\|_1^2$$
$$\leq F(w_t) + \eta_t \nabla F(w_t) \cdot (v_t - w_t) + \tfrac{1}{2}\eta_t^2 \beta D^2.$$

Next, due to the minimality of $v_t$, notice that for any $w^* \in W$,

$$\nabla F(w_t) \cdot (v_t - w_t) = g_t \cdot (v_t - w_t) + (\nabla F(w_t) - g_t) \cdot (v_t - w_t)$$
$$\leq g_t \cdot (w^* - w_t) + (\nabla F(w_t) - g_t) \cdot (v_t - w_t)$$
$$= \nabla F(w_t) \cdot (w^* - w_t) + (\nabla F(w_t) - g_t) \cdot (v_t - w^*)$$
$$\leq F(w^*) - F(w_t) + D\|\nabla F(w_t) - g_t\|_\infty,$$

---
[5]The claim of this particular theorem holds in fact for any pair of dual norms, but for consistency, is stated and proved here only for the $L_1$-$L_\infty$ case.

where the final inequality follows from convexity and the diameter bound. Plugging into the inequality and rearranging, we obtain

$$F(w_{t+1}) - F(w^*) \le (1 - \eta_t)(F(w_t) - F(w^*)) + \eta_t D \|\nabla F(w_t) - g_t\|_\infty + \tfrac{1}{2}\eta_t^2 \beta D^2.$$

Taking the expectation of both sides and using our assumption on the gradient estimates, we obtain the following progress inequality, that holds for all $t \ge 1$:

$$\mathbb{E}[F(w_{t+1}) - F(w^*)] \le (1 - \eta_t)\mathbb{E}[F(w_t) - F(w^*)] + \tfrac{1}{2}\eta_t^2(\beta D^2 + 2cD).$$

We are now ready to prove the main claim by induction on $t \ge 2$. First, plugging $t = 1$ and $\eta_1 = 1$ into the progress inequality we have that $\mathbb{E}[F(w_2) - F(w^*)] \le \tfrac{1}{2}(\beta D^2 + 2cD)$, that is, the claim holds for $t = 2$. For the induction step, let us assume that $\mathbb{E}[F(w_t) - F(w^*)] \le (\beta D^2 + 2cD)/t$ for some $t \ge 2$. Then by the progress inequality:

$$\begin{aligned}
\mathbb{E}[F(w_{t+1}) - F(w^*)] &\le (1 - \eta_t)\mathbb{E}[F(w_t) - F(w^*)] + \tfrac{1}{2}\eta_t^2(\beta D^2 + 2cD) \\
&= \left(1 - \frac{1}{t}\right)\frac{\beta D^2 + 2cD}{t} + \frac{\beta D^2 + 2cD}{2t^2} \\
&\le \frac{\beta D^2 + 2cD}{t}.
\end{aligned}$$

This concludes the proof in expectation. The second claim regarding convergence with high probability follows from the same arguments together with a union bound. $\square$

The next lemma analyzes the variance-reduced (SPIDER) gradient estimates used in Algorithm 3.

**Lemma 7.** *If we set the batch sizes as defined in Eq. (7), then for all t and $\delta > 0$:*

$$\mathbb{E}\|g_t - \nabla F(w_t)\|_\infty \le \frac{12\beta D}{t}\sqrt{\log d}, \quad \text{and} \quad \Pr\left(\|g_t - \nabla F(w_t)\|_\infty > \frac{12\beta D}{t}\sqrt{\log \frac{d}{\delta}}\right) \le \delta. \quad (8)$$

*Proof.* The proof relies on standard properties of sub-Gaussian random variables, all of which can be found in, e.g., [27]. Let $Z_{t,j} = (g_t - \nabla F(w_t))_j$ be the $j$'th coordinate of the error at step $t$. Note that $Z_{t,j}$ is zero-mean. We will show that $Z_{t,j}$ is also sub-Gaussian with parameter $\sigma_t = 6\beta D/t$. The claim will then follow since $\|g_t - \nabla F(w_t)\|_\infty$ is a maximum of $2d$ zero-mean sub-Gaussians with parameter $\sigma_t$, which implies Eq. (8).

First, consider $t$ of the form $t = \tau_k$ and any coordinate $j \in [d]$. The claim follows in this case from variance reduction by mini-batching: condition on randomness before step $t$; since all gradients are bounded in $L_\infty$ norm by $G$, we have by Hoeffding's Lemma that $(\nabla f(w_t, z_i) - \nabla F(w_t))_j$ are independent zero-mean $G$-sub-Gaussians RVs (for $i = 1, \dots, b_t$), thus their mean $Z_{t,j} = (g_t - \nabla F(w_t))_j$ is zero-mean sub-Gaussian with parameter $\sigma_t = G/\sqrt{b_t} = \beta D/t$.

Next, consider any other round such that $\tau_k < t < \tau_{k+1}$ and any coordinate $j \in [d]$. In this case, the claim will follow from accumulation of variance starting from the reset step $\tau_k$. Note that due to smoothness,

$$\begin{aligned}
|((\nabla f(w_t, z_i) &- \nabla F(w_t))_j - (\nabla f(w_{t-1}, z_i) - \nabla F(w_{t-1}))_j| \\
&\le |(\nabla f(w_t, z_i) - \nabla f(w_{t-1}, z_i))_j| + |(\nabla F(w_t) - \nabla F(w_{t-1}))_j| \\
&\le \|\nabla f(w_t, z_i) - \nabla f(w_{t-1}, z_i)\|_\infty + \|\nabla F(w_t) - \nabla F(w_{t-1})\|_\infty \\
&\le 2\beta\|w_t - w_{t-1}\|_1 \\
&\le 2\beta D\eta_{t-1}.
\end{aligned}$$

Thus, as before, the random variable $Z_{t,j} - Z_{t-1,j} = (g_t - \nabla F(w_t))_j - (g_{t-1} - \nabla F(w_{t-1}))_j$ is a zero-mean sub-Gaussian with parameter $2\beta D\eta_{t-1}/\sqrt{b_t}$, conditioned on randomness before step $t$. By the martingale-summation property of sub-Gaussian RVs, we then obtain that $Z_{t,j}$ is zero-mean

sub-Gaussian with parameter $\sigma_t$, where

$$
\begin{aligned}
\sigma_t^2 &= \frac{G^2}{b_{\tau_k}} + \sum_{s=\tau_k+1}^{t} \frac{4\beta^2 D^2 \eta_{s-1}^2}{b_s} \\
&= \frac{\beta^2 D^2}{\tau_k^2} + 4\beta^2 D^2 \sum_{s=\tau_k+1}^{t} \frac{1}{s(s-1)^2} \\
&\leq \frac{\beta^2 D^2}{\tau_k^2} + 8\beta^2 D^2 \sum_{s=\tau_k+1}^{t} \left( \frac{1}{(s-1)^2} - \frac{1}{s^2} \right) \\
&\leq \frac{\beta^2 D^2}{\tau_k^2} + \frac{8\beta^2 D^2}{\tau_k^2} \\
&\leq \frac{36\beta^2 D^2}{t^2}
\end{aligned}
$$

where the final inequality uses $\tau_k \geq t/2$. Therefore, we deduce that $\sigma_t \leq 6\beta D/t$. $\qquad\square$

We are now ready to prove Theorem 3.

*Proof of Theorem 3.* The theorem follows directly from Lemmas 6 and 7 and plugging in the specified parameters. The number of steps for convergence to within a given $\epsilon > 0$ is therefore $t = O((\beta D^2/\epsilon)\sqrt{\log(d/\delta)})$, which is also the number of LOO calls. For the total sample complexity, observe that the contribution of rounds $t \notin \{\tau_k\}_{k\geq 1}$ is bounded by $\sum_{s=1}^{t} s = O(t^2)$, while the contribution of rounds $t \in \{\tau_k\}_{k\geq 1}$ is at most $O((G/\beta D)^2 t^2)$. $\qquad\square$

