# OpenReview forum: "Fast Rates for Bandit PAC Multiclass Classification"
_NeurIPS.cc/2024/Conference — NeurIPS 2024 poster_

### Official Review · Reviewer_Lcb5 · 2024-06-24

**Soundness:** 4
**Presentation:** 4
**Contribution:** 3
**Rating:** 7
**Confidence:** 2

**Summary:**

This manuscript deals with Multiclass (K labels) PAC Classification a partial monitoring scheme, as introduced by Daniely et al. ('11). The complexity for $(\epsilon,\delta)$-PAC of a naive uniform sampling algorithm would be $K/\epsilon^2 \log (|\mathcal{H}| / \delta) \big)$ where $\mathcal{H}$ is a finite family of classifiers. This manuscript introduces a two-step procedure that achieves the nearly optimal bound $(poly(K) + 1/\epsilon^2)\log(1/\delta)$. This bounds is optimal even in the simpler setting with complete feedback. Then, the authors extend the procedure to the infinite class with finite Natarajan dimension.

**Strengths:**

1) The manuscript solves an open question in multi-class learning literature under partial monitoring. Although I am not an expert in the field, I feel that the idea a weighed ERM in the second step of the algorithm and to connect that to the guarantee of the SPIDER gradient estimates for Franck-Wolf algorithm.

2) The manuscript is very well written and the authors manage to convey well the main ideas behind their procedures. The proofs are clear.

**Weaknesses:**

None.

**Questions:**

Line 177: What is J_{x,y}?

Could the procedure allows for a better dependency with respect to $K$?

**Limitations:**

Yes.

---

> ### Author Rebuttal · Authors · 2024-08-05
>
> We thank the reviewer for the time and effort. Please see our comment below to the specified questions.
>
> * “Line 177: What is J_{x,y}?”
>
> This is a typo and should be replaced by $r_{x,y}$, this will be fixed in the final version.  Thanks for catching!
>
> * “Could the procedure allows for a better dependency with respect to $K$?”
>
> We believe that the polynomial dependence on $K$ in the regret bound could be reduced, but we are currently unsure how exactly this can be done and what is the correct optimal dependence.

---

> > ### Comment · Reviewer_Lcb5 · 2024-08-07
> >
> > Thank you for your response.

---

### Official Review · Reviewer_Wz7L · 2024-07-05

**Soundness:** 4
**Presentation:** 3
**Contribution:** 3
**Rating:** 7
**Confidence:** 3

**Summary:**

The authors present a novel algorithm to solve the epislon-delta PAC bandit multi-class classification problem.

For a finite hypothesis class, they give an algorithm with sample complexity O(poly(K) + 1/epsilon^2) that improves on the O(K/epsilon^2). They also similarly show that for a possibly infinite class they can also improve the state-of-the-art.

**Strengths:**

The paper solves an existing know issue without introduction too many new assumptions. They show the bounds match the optimal rate in the full-information regime. Their use of the Frank-Wolfe algorithm is also interesting.

**Weaknesses:**

The improvement is shown by moving from O(K/epsilon^2) to O(poly(K) + 1/epsilon^2). The constants here are on the order of K^8 and the time horizon is on the order of K^4. This means that if K is relative large, you mind end up doing worse than the previous bounds which limits the scope of impact of this work.

**Questions:**

What would be a scenario where this bound performs better than the previous work, although theoretically this is better?

**Limitations:**

Yes

---

> ### Author Rebuttal · Authors · 2024-08-05
>
> We thank the reviewer for the time and effort. Please see our comment below to the specified weaknesses and questions.
>
> * “...This means that if K is relative large, you mind end up doing worse than the previous bounds which limits the scope of impact of this work.”
>
> You are correct in that our current bounds improve upon previous results in regimes where $K$ is not too large relative to $1 / \epsilon$. However, we believe that removing polynomial dependencies in $K$ from the main $1/\epsilon^2$ term is a crucial and significant step towards a comprehensive understanding of the bandit multiclass problem.
>
> * “What would be a scenario where this bound performs better than the previous work, although theoretically this is better?”
>
> Since this is a purely theoretical study, we do not focus on empirical scenarios in which our algorithm may outperform existing ones. Our results formally show that if we aim to find an $\epsilon$-optimal policy where $\epsilon \ll 1 / K^4$, then our suggested algorithm uses considerably less samples than known algorithms which require $K / \epsilon^2$ samples.

---

> > ### Comment · Reviewer_Wz7L · 2024-08-14
> > **Response to rebuttal**
> >
> > Thank you for your response. I maintain my rating of accept

---

### Official Review · Reviewer_m8fV · 2024-07-06

**Soundness:** 4
**Presentation:** 4
**Contribution:** 4
**Rating:** 7
**Confidence:** 4

**Summary:**

This paper studies bandit multiclass classification in the agnostic PAC setting. For both finite and infinite hypothesis classes, they provide efficient learning algorithms (assuming access to a weighted ERM oracle) whose sample complexity significantly improves upon the previous best known rate of $\frac{K}{\epsilon^2}$. As a corollary, they show that unlike the realizable PAC setting, there is no price of bandit feedback in the agnostic PAC setting. The main algorithmic ideas involve  a stochastic optimization technique for efficiently recovering a low-variance exploration distribution over the hypothesis class.

**Strengths:**

- This paper is very well written and easy to follow
- The technical contributions are solid, results are interesting and surprising, and proof techniques are non-trivial. In particular, the fact that there is no price of bandit feedback in the agnostic PAC setting is interesting and not obvious apriori
- This paper resolves an open question posed by Daniely, Sabato, Ben-David, and Shalev-Shwartz (2011)
- Overall, I feel that this paper is a nice contribution to learning theory that furthers our understanding of bandit multiclass learnability

**Weaknesses:**

There are no major weakness. That said, I think it would be nice to have:

- a more detailed discussion of the realizable bandit PAC setting in the main text summarizing the known rates and algorithmic techniques
- some concrete practical scenarios where the bandit agnostic PAC setting is realistic
- discussion of the assumption of finite label spaces and some thoughts on the right characterization of bandit PAC learnability and sample complexity when $K$ goes to infinity
- some intuition (beyond the fact that the proof works) about why a separation occurs between the realizable and agnostic settings with regards to the price of bandit feedback

**Questions:**

- What are the known lower bounds (if any) in the bandit agnostic PAC setting? Is a poly factor of $K$ unavoidable?
- When $K$ is finite, bandit PAC learnability is equivalent to PAC learnability. Presumably, this is not the case when $K$ is allowed to be infinite. Do you have any insight on what the right characterization of bandit PAC learnability might be in this case?

**Limitations:**

No limitations.

---

> ### Author Rebuttal · Authors · 2024-08-05
>
> We thank the reviewer for the time and effort. Please see our comment below to the specified questions.
>
> * “What are the known lower bounds (if any) in the bandit agnostic PAC setting? Is a poly factor of 𝐾 unavoidable?”
>
> We can prove a simple lower bound of $K / \epsilon + 1 / \epsilon^2$ for bandit multiclass classification as follows: The $1 / \epsilon^2$ term arises from a standard argument for two labels with probabilities $1/2 + \epsilon$ and $1/2 - \epsilon$ (this is the full information lower bound). The $K / \epsilon$ lower bound holds for an instance with two examples $x_1$ and $x_2$ and $K$ labels, where the probability of sampling $x_1$ is $2 \epsilon$ and $x_2$ has probability $1-2 \epsilon$ (each example has a unique unknown correct label). A PAC learner for this instance must correctly predict the label of $x_1$, and to do that it needs at least $\Omega(K)$ samples of $x_1$, which takes a total of $\Omega(K / \epsilon)$ samples. We believe that the polynomial dependence on $K$ in our upper bound can be further improved to decrease the gap from the lower bound described above, but we are unsure how this can be done with our current techniques.
>
> * “...Do you have any insight on what the right characterization of bandit PAC learnability might be in this case?”
>
> Good question. Yes, there is a relatively simple characterization: a class H is learnable if and only if the following holds: (i) it has finite Natarajan dimension, (ii) there is a finite bound $K$ such that for every point $x$, the set {$h(x) : h \in H$} has at most $K$ distinct labels. Item (i) is clearly necessary, and it is relatively simple to prove that Item (ii) is necessary (it is necessary already for learnability in the realizable-case). For sufficiency, note that if Item (ii) holds then the class $H$ is effectively equivalent to a class over $K$ labels in total and therefore can be learned when its Natarajan dimension is finite, as follows from Theorem 2 in our work. We will add this to the final version of our paper, thanks for pointing this out!

---

> > ### Comment · Reviewer_m8fV · 2024-08-07
> >
> > Thanks for the response.

---

### Official Review · Reviewer_xcWH · 2024-07-09

**Soundness:** 4
**Presentation:** 4
**Contribution:** 3
**Rating:** 6
**Confidence:** 4

**Summary:**

This paper studies the problem of multiclass classification with bandit feedback, where one only receives information on whether the prediction is correct or incorrect without revealing the actual label. This can be viewed as a very specific case of the well-known contextual multi-armed bandits, where the cost vector contains a single value of $0$ and one's elsewhere. The paper shows that unlike the full-power contextual multi-armed bandits, where the sample complexity scales as $\Omega(K/\epsilon^2)$, this restricted version has a sample complexity that scales as $O(K^9+ 1/\epsilon^2)$. Moreover, their algorithm is computationally efficient given an ERM oracle.

**Strengths:**

This paper addresses a natural and fundamental problem that has attracted much attention in the literature. It introduces several original techniques, such as finding exploration distributions via a log-barrier potential as in (2). I also find the implementation of the Stochastic Frank-Wolfe algorithm using an ERM oracle to be quite interesting. The paper is quite clean and easy to read. I believe it is worth being published in NeurIPS and would have impact within the learning theory community.

**Weaknesses:**

I do not see any significant weaknesses in the paper. This is a neat, pure theory paper that addresses an interesting problem, at least within the community.

I have a few minor remarks as follows:

1. It appears to me that the $K^9$ dependency mainly arises from the Stochastic Frank-Wolfe algorithm. Can this be improved? Perhaps with a computationally inefficient algorithm?

2. The paper claims to resolve an open problem from Daniely, Sabato, Ben-David, and Shalev-Shwartz (2011). This sounds a bit overselling, in my opinion. After browsing the cited paper, I did not find any statement of this problem, particularly regarding the dependency on $1/\epsilon$.

3. Can the techniques developed in this paper be used in more general bandit settings to yield improved bounds?

4. The $J_{x,y}(h)$ after Eq (3) must be $r_{x,y}(h)$.

5. It appears to me that your technical approach is quite similar to that of [12]. Can the authors comment on the similarities and differences with that work?

**Questions:**

See above.

---

> ### Author Rebuttal · Authors · 2024-08-05
>
> We thank the reviewer for the time and effort. Please see our comment below to the specified weaknesses and questions.
>
> * “...Can this be improved? Perhaps with a computationally inefficient algorithm?”
>
> We believe that the sample complexity (specifically, the polynomial dependence on $K$) could be reduced even with a computationally efficient algorithm, but we are not sure how to do that with our current approach.
>
> * “The paper claims to resolve an open problem from Daniely, Sabato, Ben-David, and Shalev-Shwartz (2011)...”
>
> Specifically, we refer to the paragraph titled “The price of bandit information in the batch model” in section 4 of Daniely et al. (2011), where the authors note that “...it would be interesting to find a more tight characterization of the sample complexity in the bandit setting” while observing that the known multiplicative gap was linear (up to log factors) with the size of the label space. Since our results establish a sample complexity bound that matches (up to log factors) the full information bound for small values of $\epsilon$, we view our results as a resolution of this question.
>
> * “Can the techniques developed in this paper be used in more general bandit settings to yield improved bounds?”
>
> In the more general contextual bandit setup where there is no presumed structure on the reward functions, the best sample complexity achievable is of the form $K / \epsilon^2$ which can be obtained by the trivial algorithm which pulls actions uniformly at random and returns the policy with the best empirical reward. For the online objective of regret minimization, we believe that variants of our approach can be used to obtain the optimal $\sqrt{KT}$ regret in contextual bandits with a computationally efficient algorithm which is perhaps simpler than the previous approaches of Dudik et al. (2011) and Agrawal et al. (2014), but we did not attempt to work this out in full detail.
>
> * “The $J_{x,y}(h)$ after Eq (3) must be $r_{x,y}(h)$.”
>
> You are correct, this is a typo that will be fixed in the final version.  Thanks for catching!
>
> * “It appears to me that your technical approach is quite similar to that of [12]. Can the authors comment on the similarities and differences with that work?”
>
> The work of [12] considers the more general contextual bandit setting with the online objective of regret minimization. Their approach essentially amounts to adaptively computing low-variance sampling distributions which also induce high estimated reward in order to minimize regret over time. Our approach is similar in the sense that we also aim to compute a low-variance sampling distribution with which to estimate rewards of hypotheses, but in the setting we consider there is a special structure on the rewards (namely, a “sparse” one-hot structure) which allows us to compute such a low-variance distribution in time complexity that is polynomial in $K$, and then use it to estimate the rewards of all hypotheses uniformly with only $K / \epsilon + 1 / \epsilon^2$ samples. This approach would not work for regret minimization, since the initial phase alone would incur linear regret. We also add that our algorithm makes use of a stochastic Frank-Wolfe procedure in order to efficiently compute the low-variance sampling distribution, an approach which highly differs from the optimization scheme employed in each round in [12].

---

> ### Comment · Reviewer_xcWH · 2024-08-07
>
> Thank you for the response. I maintain my current rating, favoring the acceptance of the paper.

---

### Official Review · Reviewer_TyCi · 2024-07-13

**Soundness:** 3
**Presentation:** 3
**Contribution:** 3
**Rating:** 7
**Confidence:** 3

**Summary:**

This paper considers the problem of bandit multi class classification, where the learner only receives the true label if their prediction was correct. That is at time $t$ the learner receives a training sample $x_t$ with unknown label $y_t$. They then predict one of $K$ labels and are told if their prediction is correct. This problem is well studied in the case of regret minimisation. The authors consider a fixed confidence setting where the learner must be PAC$(\epsilon,\delta)$ in terms of classification error when compared to the best classifier of some hypothesis class. In addition to receiving samples the learner has access to a ERM oracle on the hypothesis class.

For the case of a finite hypothesis class the authors provide an algorithm with upper bound of the order $(K^9 + \epsilon)\log(\frac{N}{\delta})$ where $N$ is the size of the hypothesis class. For hypothesis class of potentially infinite size, but finite Natarajan dimension $d_N$, the authors provide an algorithm with upper bound $(K^9 + \epsilon)d_N\log(\frac{N}{\delta})$.

**Strengths:**

Bandit multi class classification has been studied, mainly in the case of regret minimisation. I think its interesting to consider the problem in such a PAC setting. The question of whether one should pay $K/\epsilon^2$ under bandit feedback, where one can achieve a rate of $1/\epsilon^2$ in the full information case, appears to have been a question of interest for some time.

**Weaknesses:**

As mentioned by the authors, in practical settings the number of labels may be large and the $K^9$ term is significant. While one should keep in mind, that this work appears to be the first to get rates not dependent on $K$ as $\epsilon \rightarrow 0$, it would be nice for the authors to discuss how future works could reduce the degree of the poly dependence upon $K$.

It would also be nice to have more discussion when comparing to contextual bandits. The authors direct the reader to [13] for a more detailed comparison, however from what I can tell they consider solely regret minimisation therein and not a PAC setting.

I believe the paper would benefit from a conclusion and perspectives on future work in the main text.

**Questions:**

Do the authors think there is a relation to contextual bandits in the PAC setting e.g. "Instance-optimal PAC Algorithms for Contextual Bandits" Li et al. ?

Have the authors considered instance dependent bounds, specifically for a finite hypothesis class?

**Limitations:**

No concern.

---

> ### Author Rebuttal · Authors · 2024-08-05
>
> We thank the reviewer for the time and effort. Please see our comment below to the specified weaknesses and questions.
>
> * “...it would be nice for the authors to discuss how future works could reduce the degree of the poly dependence upon $K$.”
>
> We believe that the polynomial dependence on $K$ could be reduced, but we are currently not sure how this can be done with our current techniques. One possibility is to employ an adaptive approach which combines estimated reward maximization together with lowering the variance of the sampling distribution over time. We will incorporate a short discussion in the revision, thanks for this suggestion.
>
> * “It would also be nice to have more discussion when comparing to contextual bandits…”
>
> The vast majority of previous works on contextual bandits study the online objective of regret minimization. Our setting of bandit PAC multiclass classification most closely resembles a contextual bandit setup with a different objective of finding a nearly optimal policy with as few samples as possible.
>
> * “I believe the paper would benefit from a conclusion and perspectives on future work in the main text.”
>
> Agreed - we will add a conclusion section in the revision, incorporating a discussion of the two points you suggested earlier.
>
> * “Do the authors think there is a relation to contextual bandits in the PAC setting e.g. "Instance-optimal PAC Algorithms for Contextual Bandits" Li et al. ?”
>
> Thank you for bringing this paper into our attention; we will make sure to cite and discuss it in subsequent versions of our paper. The work of Li et al. considers the PAC variant of contextual bandits, where no structural assumptions are made apriori on the reward functions. In this setting, they prove instance dependent sample complexity bounds, from which worst-case bounds of the form $ \approx K / \epsilon^2$ can be inferred. To our understanding, there is no direct relation between instance dependent bounds and the single-label classification setting which we consider, as even for the simple case of multi-armed bandits, the standard sample complexity bound of the form $\sum_i (1 / \Delta_i)$ does not suffice to obtain an improved result for sparse rewards. To obtain adaptivity to sparsity, the bounds must actually be variance-dependent (rather than gap-dependent), which to our understanding is not addressed in the work of Li et al.
>
> * “Have the authors considered instance dependent bounds, specifically for a finite hypothesis class?”
>
> We have not previously considered instance-dependent (that is, gap-dependent) sample complexity bounds, as obtaining improved instance-independent bounds, regardless of sparsity, is in itself a highly nontrivial problem. Obtaining such bounds in the multiclass setting is a fantastic question for future research.

---

### Decision · Program_Chairs · 2024-09-25

**Decision:**

Accept (poster)

**Comment:**

This paper gives a new rate of PAC bandit multiclass classification by improving the known O(K/epsilon^2) sample complexity to O( poly(K) + 1/epsilon^2 ). As a consequence, the price of bandit feedback in agnostic PAC multiclass classification (when epsilon is small) is settled, resolving an open problem in (Daniely et al, 2011), which is a constant, in contrast with the realizable setting. The reviewers find the result and algorithmic approach interesting and think this paper should be accepted.

In the final version, we encourage the authors to incorporate discussions in e.g. lower bounds, and implications to infinite label spaces.